# Cross-Modal Coherence-Enhanced Feedback Prompting for News Captioning

Submission Id: 4313*

## ABSTRACT

News Captioning involves generating the descriptions for news images based on the detailed content of related news articles. Given that these articles often contain extensive information not directly related to the image, captions may end up misaligned with the visual content. To mitigate this issue, we propose the novel cross-modal coherence-enhanced feedback prompting method to clarify the crucial elements that align closely with the visual content for news captioning. Specifically, we first adapt CLIP to develop a news-specific image-text matching module, enriched with insights from language model MPNet using a matching-score comparative loss, which facilitates effective cross-modal knowledge distillation. This module enhance the coherence between images and each news sentences via rating confidence. Then, we design confidence-aware prompts to fine-tune LLaVA model with by LoRa strategy, focusing on essential details in extensive articles. Lastly, we evaluate the generated news caption with refined CLIP, constructing confidence-feedback prompts to further enhance LLaVA through feedback learning, which iteratively refine captions to improve its accuracy. Extensive experiments conduct on two public datasets, GoodNews and NYTimes800k, have validated the effectiveness of our method.

## CCS CONCEPTS

• **Computing methodologies** → **Scene understanding**.

## KEYWORDS

News Captioning, Coherence-Enhanced, Feedback Learning.

## 1 INTRODUCTION

In modern news dissemination, images gradually serve a crucial function, significantly enriching news content. This motivates the emergence of news captioning task [28, 40], which automatically generates descriptive text that matches news image content. As illustrated in Fig. 1, different from general image captioning [14, 37], news captioning not only identifies visual elements within news images, but also delves into the associated background knowledge and named entities in news articles (e.g. people's names, organizations, locations), making the caption more rich and specific.

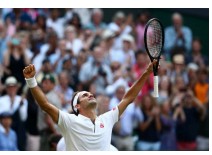

**General Captioner:** Tennis player celebrating a victory with a raised racket.
**News Captioner:** Roger Federer is seeking his 21st Grand Slam singles title and ninth championship at Wimbledon.

**Article:** How Federer Beat Nadal to Advance to the Wimbledon Final. WIMBLEDON, England Unlike Roger Federer and Rafael Nadal' duel into the twilight on Centre Court in 2008, their match Friday was not a contender for the greatest match of all time. But it had its moments, both transcendent and surprising, and it also had a different finish ...

**Figure 1: Comparison between general image captioning and news captioning. News captioner can generate fine-grained descriptions that include specific events and named entities.**

Early news captioning task primarily relied on template-based methods [2, 22], which aim to construct a caption template containing placeholders that are then populated with the appropriate named entities. To further enhance the flexibility and enrich the content of generated news captions, recent researches begin to shift towards fine-tuning the pre-trained model based on multi-modal prompts [28, 35]. Although the current methods have shown their superiority in news captioning, most of them ignore a problem that long context in news articles contain a substantial amount of named entities irrelevant to the news image, which introduces additional noise for news captioner. The presence of noise will interfere the ability of news captioner to accurately capture image-relevant key details from the news article, resulting in generating unrelated captions. As shown in Fig. 2(a), information noise cause the model to locate the false news context, thereby mismatching the wrong named entities '*Louis Lester Band*' as the subject of output caption.

An intuitive approach to addressing this challenge is directly extracting critical sentences from news article as input of captioner [45]. However, this strategy presents two significant limitations: current pre-trained image-text matching models (i.e. CLIP [21]) lack the capability for precise alignment in news contents, which is difficult to clarify irrelevant information. Secondly, the quality of the generated captions substantially depends on the content from news article. Simply filtering out the input contents will risk the loss of contextual details, and thus potentially diminishing the general comprehensiveness of news article. In this case, there is an absence of a feedback optimization mechanism within news captioners to mitigate the impact of irrelevant content deriving from input noise.

To solve the above drawbacks, we propose the novel cross-modal coherence-enhanced feedback prompting method for news captioning. As shown in Fig. 2 (b), to adapt multi-modal pre-trained CLIP model specifically for the news domain, we first implement a cross-modal knowledge distillation strategy. Concretely, we adopt

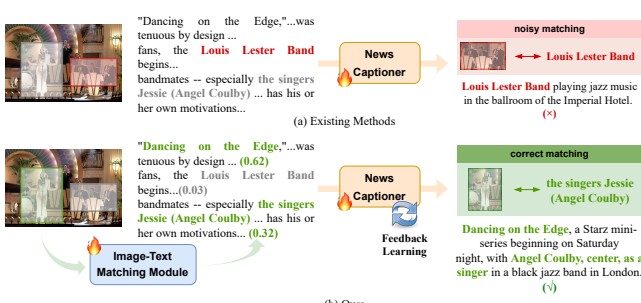

**Figure 2: Comparison between existing methods and ours. Compared to the existing methods, we utilizes an image-text matching module and a feedback learning strategy to respectively address noise interference in the input and output processes, achieving more accurate news captions.**

the pre-trained large language model MPNet [25] as the teacher network to score each news sentence with ground truth caption for obtaining alignment knowledge. These aligned scores are then integrated into a pre-trained CLIP model, serving as the student network, to perform fine-tuning through matching-score comparative loss function. This process facilitates cross-modal knowledge distillation and refines CLIP into a domain-specific image-text matching module, significantly improving its comprehension for the coherence between each news sentence and corresponding images. The refined CLIP is employed to accurately assess the alignment confidence scores between text and image of news. Subsequently, we adopt LLaVA model [16] as the news captioner in our method. To empower the LLaVA model with enhanced discernment of news article contexts, we design a confidence-aware prompting mechanism to refine its multi-modal capabilities via the LoRa optimization strategy [10], thereby diminishing the impact of input noise based on aligned confidence score. Furthermore, to mitigate the generation of irrelevant captions resultant from input noise, we conduct evaluations of the generated news captions using the refined CLIP model, leading to the introduction of a feedback learning mechanism. This mechanism establishes a confidence-feedback prompt to enable the LLaVA model autonomously refining its output through successive iterations, which resolves the issue of generating irrelevant descriptions caused by input noise and achieve more accurate news caption.

We conduct extensive experiments on two publicly news captioning datasets: GoodNews [2] and NYTimes800k [28]. The results demonstrate that our method achieves state-of-the-art performance on most metrics across both datasets. In summary, the key contributions of this work are as follows:

(1) We propose the novel cross-modal coherence-enhanced feedback prompting method for news captioning, which effectively mitigates the negative impact of irrelevant information from news article for captioner.

(2) We refine the CLIP model through cross-modal knowledge distillation strategy with language model MPNet, significantly enhancing its news-specific image-text matching ability. We further design a feedback learning mechanism to

enable LLaVA model iteratively refining news captions with a confidence-aware prompting, ensuring the improvements in caption accuracy and relevance.

(3) Our research involves a comprehensive evaluation on two news captioning datasets, GoodNews and NYTimes800k, to demonstrate its superiority. We achieve the best results in most general image caption metrics, including measures of named entity accuracy and recall.

## 2 RELATED WORK

### 2.1 General Image Captioning

Image captioning task has made significant progress in recent years [3, 5, 13, 17, 30–32, 41]. Researchers begin to expand enriched image captions more in line with human preferences by supplementing knowledge of visual entities, enhancing visual semantics, improving sentence generation strategies and so on. Yang et al. [36] integrate language inductive bias into image captioner to generate more human-like image captions. Zha et al. [38] propose a context-aware visual policy network that improves image caption generation by considering visual context. Hosseinzadeh et al. [8] propose an approach to improve image change caption by introducing composite query image retrieval as an auxiliary task. Li et al. [15] use extracted style phrases to generate rich and diverse descriptions for images. Zhou et al. [46] propose a semi autoregressive model that preserves autoregressive properties globally while achieving parallel word generation locally. Compared to general image captioning, our method is capable of generating descriptions that include fine-grained information such as named entities, based on the given image-article pairs.

### 2.2 News Captioning

The news captioning task [2, 9, 28, 45] aims to generate textual descriptions that capture specific events or named entities in images embedded in news articles. Biten et al. [2] adopted a two-stage news caption generation method, which first generates template captions with placeholders, and then selects appropriate named entities from the contextual information of news articles to replace the placeholders in the template captions. Tran et al. [28] propose an end-to-end model that generates descriptions containing rare words by using a byte-pair-encoding transformer language model. Yang et al. [35] follow the journalist's caption guidelines, predict template components (e.g., who, when, where) based on image-article pairs, and then generate news descriptions based on the guidance of these components. Zhang et al. [40] achieve global entity perception in news captions by combining pre-trained models CLIP and BART and introducing multi-modal entity prompts. Qu et al. [20] propose optimizing the embedding representation of people's names and using CLIP to retrieve sentences that are semantically similar to images, in order to simulate the process of humans searching for relevant article information based on images to generate news captions. Although existing methods have considered weakening the input noise from long news articles by retrieving more granular news sentences from the articles, their method of filtering relevant sentences loses certain semantic information while filtering out noise. Moreover, they have not taken into

account the issue of output noise caused by input noise. In comparison, our method utilizes image-text matching confidence-aware prompts to effectively reduce the interference of information noise without losing semantic information. Additionally, by employing feedback learning, we further minimize output noise. Therefore, our approach is more suitable for the news captioning task.

## 2.3 Prompt Learning for Multi-modal Pre-trained Model

Prompt learning as an effective strategy to enhance the performance of pre-trained models, has been widely applied in multimodal downstream task [7, 11, 24, 26, 33, 34, 43, 44]. Khattak et al. [12] use multi-modal cues to fine-tune CLIP, aligning visual and linguistic representations. Zhao et al. [39] propose the multiprompts ReID framework, which utilizes a multi-prompts learning strategy to generate diverse sentences and improve the accuracy of person re-identification. Chowdhury et al. [4] propose APoLLo, a multimodal method that combines adapters and prompt learning, aimed at improving the generalization ability of visual-language pre-trained models in few-shot setting. Zhang et al. [42] improve the generalization ability of CLIP by creating a visual concept cache. Inspired by [24], we use the image-text matching module to calculate the matching confidence between images and news sentences, and then use matching confidence prompts to enhance the multimodal pre-trained model LLaVA's ability to perceive key contexts in news articles, thereby generating more accurate descriptions.

## 3 APPROACH

### 3.1 Overview

The objective of news captioning task is to generate a caption, denoted as $y_{news}$, that accurately reflects the content of an image based on the corresponding image-article pair $(I, A)$. To mitigate the adverse effects of irrelevant information from news article, we propose the novel cross-modal coherence-enhanced feedback prompting method for news captioning. As shown in Fig. 3, our approach comprises two principal components: 1) **Cross-Modal Knowledge Distillation**: This component addresses information noise at the input phase. Through knowledge distillation, we facilitate the transfer of textual alignment expertise from the pre-trained language model MPNet [25] to the multi-modal model CLIP, thereby establishing a news-domain-specific image-text matching module. This module enhance the coherence between each news sentences and images via rating their confidence. 2) **Match-Confidence Prompt Enhancement**: Based on the confidence score, we further design the second component to handle the output noise of news captioner. We adopt a multi-modal pre-trained model LLaVA as news captioner and propose a feedback learning mechanism, fine-tuning the LLaVA model to iteratively generate news caption using confidence-feedback prompt, which ultimately improves the quality of generated news captions.

### 3.2 Cross-Modal Knowledge Distillation

For the input of news captioner, we require that each sentence of news article contains the degree of relevance to the image, just like the example shown in Fig. 2 (b). This kind of relevance may involve

complex narratives and specialized vocabulary that go beyond simple visual-textual matching. The general CLIP model is trained on a wide variety of internet-sourced image-text pairs, which may not adequately represent the specific characteristics of news content. To close the semantic gap, we transfer the text alignment knowledge from the pre-trained language model MPNet to fine-tune the CLIP through cross-modal knowledge distillation, thereby constructing an image-text matching module applicable to the news domain.

Specifically, the datasets of news captioning provide news image $I$, the corresponding articles $A$ and ground truth news caption $c$. We first collect a set of positive image-text pairs from these news captioning datasets for contrastive learning, including both standard-positive and potential-positive samples.

**1) Standard-Positive Samples.** Each pair within the standard-positive sample set comprises an image $I$ and its corresponding ground truth caption $c$. These captions are carefully crafted by professional journalists, accurately capture the key visual information of the images and effectively convey relevant background knowledge. Therefore, to acknowledge the high relevance of these pairs, we assign a matching score of 1 to each $(c, I)$ pair, thereby constituting the standard-positive triplet set $(c, I, 1)$.

**2) Potential-Positive Samples.** Potential-positive samples are designed to broaden the CLIP model's recognition capabilities, thereby augmenting its proficiency in processing and comprehending sentences derived from actual news articles. For acquiring potential-positive samples, we adopt the advanced pre-trained language model MPNet as a teacher network and its text alignment knowledge are used to fine-tune the CLIP model through matching-score comparative loss, thereby achieving the cross-modal knowledge distillation. Specifically, the MPNet model first encode the caption $c$ and the corresponding news sentence $s_n \in A$ within the article to generate respective vector representations. Cosine similarity is then employed to derive the similarity score $x_n$ for each vector pair, facilitating an evaluation of their alignment within the semantic vector space:

$$\{C, S_n\} = \text{MPNet}(c; s_n) \ , \ \ x_n = \text{CosSim}(C, S_n) \qquad (1)$$

where $C$ represents the vector of the ground truth caption $c$; $S_n$ represents the vector of the $n$-th news sentence $s_n$ in the article; MPNet$(\cdot)$ is the text encoder and CosSim$(\cdot)$ is the cosine similarity function. We consider the ground truth caption $c$ to be a fully matched text description of the image $I$, assigning it the highest matching score of 1. Based on this premise, we further assume that the similarity score $x_n$ between a news sentence $s_n$ and the caption $c$ could serve as the matching score between the sentence and the image. So we set a pre-defined threshold $\eta$ and filter out news sentences from the article, whose matching scores $x_n$ with the image exceed this threshold. We then combine these news sentences $s_n$, the corresponding image $I$ and their matching scores $x_n$ to form potential-positive triplets $(s_n, I, x_n)$. Finally, the standard-positive sample and the potential-positive samples are merged to obtain the positive samples $P$:

$$P = \{(s, I, x)\} = \{(c, I, 1)\} \cup \{(s_n, I, x_n)|x_n > \eta\} \qquad (2)$$

where standard-positive sample $\{(c, I, 1)\}$ consists of triples, containing a ground truth caption $c$, the matching score of 1 and the corresponding image $I$. The potential-positive samples $\{(s_n, I, x_n)|x_n >$

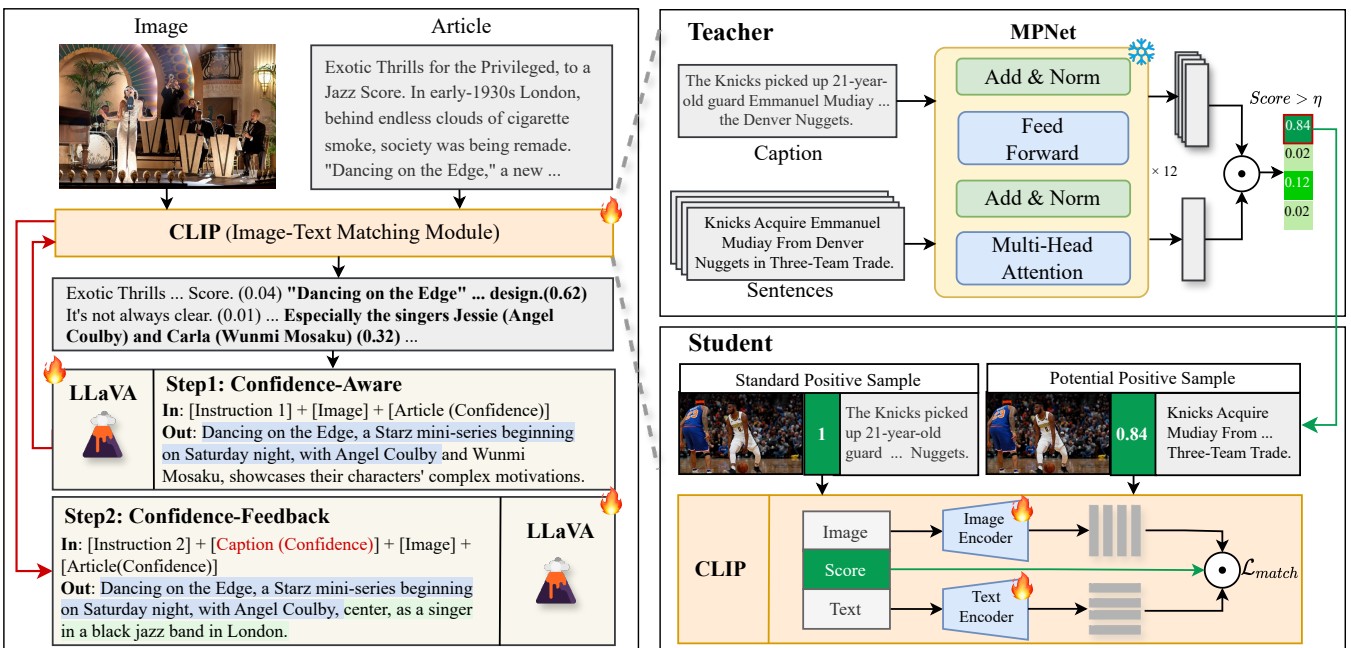

**Figure 3: Overview of the proposed method. In *Cross-Modal Knowledge Distillation*, we utilize knowledge distillation techniques to fine-tune CLIP, thereby constructing an image-text matching module suitable for the news domain. In *Match-Confidence Prompt Enhancement*, the image-text matching module is used to allocate confidence score for each sentence in the news article. Then we design a confidence-aware prompt based on confidence scores, and fine-tune LLaVA model using LoRA technology to guide the model in identifying key news context in lengthy articles, thereby generating accurate news captions. Finally, we design a feedback learning to further reduce irrelevant information in the generated captions through iterative correction.**

$\eta\}$ comprise triples, containing a news sentence $s_n$, the matching score $x_n$, and the associated image $I$.

After obtaining the positive samples $P$, we categorize news captions and sentences lacking correspondence with image $I$ as negative samples, creating a clear definition between match and mismatch cases. Then, we employ contrastive learning to refine the CLIP model, guiding it to diminish the similarity gap for positive image-text pairs and expand it for negative pairs, enhancing its discriminatory capacity. Particularly, we introduce a unique matching-score comparative loss to fine-tune CLIP model, which is engineered to variably weight each positive sample according to its computed matching score.

$$\omega = e^{\frac{\ln \eta}{\eta-1}(x-1)}, (s, I, x) \in P \tag{3}$$

$$\mathcal{L}_{match} = -\sum_{j=1}^{N} \omega_j \cdot \log\left(\frac{\exp(logit_j)}{\sum_{i=1}^{N}\exp(logit_{j,i})}\right) \cdot 1 \tag{4}$$

where $\omega$ is the matching-score weight factor; $\eta$ is the threshold for selecting potential positive samples; $N$ is the number of samples in a batch; $logit$ is the matching score between $s$ and $I$ calculated by the refined CLIP.

## 3.3 Match-Confidence Prompt Enhancement

We employ an end-to-end vision-language pre-trained model LLaVa [16] as news captioner, which incorporates a projection matrix to connect a visual encoder with a language decoder based on LLaMA [27]. It demonstrates excellent performance in multiple visual-language tasks. However, news articles often contain substantial semantic information unrelated to the image, which may introduce unnecessary noise during the input stage of the LLaVA model. Inspired by [24], we first adopt the refined CLIP model to rate the confidence of each news sentence, and then establish a confidence-aware prompt to fine-tune LLaVa model by LoRa strategy, which could make it focus on key sentences when processing lengthy articles.

Specifically, in the confidence-aware prompting enhancement stage, we utilize the refined CLIP model to compute the confidence scores between the image $I$ and each news sentence $s_n \in A$.

$$F = \text{Softmax}(\text{CLIP}_{news}(I, \{s_1, s_2, ..., s_n\})) \tag{5}$$

where $F = \{f_1, f_2, \cdots, f_n\}$ is the set of confidence scores between the image and all the corresponding news sentences. $\text{CLIP}_{news}$ denotes the refined CLIP model with news content. Then, we design a confidence-aware prompt based on these confidence scores, which consists of three components: a task instruction $p$, the news image $I$, and the article sentences $s_n \in A$ accompanied by their corresponding confidence scores $f_n \in F$. The prompt takes the following form: "***Please read the article and write a single-sentence professional news caption for the accompanying image. In the article, there is a confidence score in parentheses following each sentence,***

*which indicates the relevance of the sentence to the image.* + $I$ + ***Article:*** $s_n(f_n)$". Based on this approach, we fine-tune LLaVA to locate the key contextual information in the news article, thereby generating a more accurate news description:

$$\theta' = \arg\min_{\theta} L[y_{news}, \text{LLaVA}(p, I, s_n, f_n | \theta)] \quad (6)$$

where $L$ is the loss function used for fine-tuning; $\theta$ is the parameter of LLaVA. Here, we denote $y'_{news}$ as the generated the news caption by fine-tuned LLaVA model.

However, the input news sentences with low correlation score may also result in potential biases for news captioner to generate descriptions mismatched with the image. We further propose an innovative feedback learning mechanism to mitigate the adverse effects of input noise on output caption. This mechanism optimizes the generated news caption $y'_{news}$ through a confidence-feedback prompt strategy. Specifically, we first employ the refined CLIP model to compute the confidence score between the image $I$ and the initially generated news description $y'$, reflecting the consistency level between the description and the news image, providing a quantitative feedback signal to the model:

$$\hat{f} = \text{Norm}(\text{CLIP}_{news}(I, y')) \quad (7)$$

where $\hat{f}$ is the obtained confidence score; and $\text{Norm}(\cdot)$ denotes a normalization function. Then, we regard samples with scores below a threshold $\mu$ as unqualified and further design a confidence-feedback prompt based on this confidence score. The new prompt framework comprises four components: a task instruction $p'$, the news image $I$, the initial description $y'_{news}$ with its confidence score $\hat{f}$, and the article sentences $s_n$ accompanied by their confidence scores $f_n$. The prompt takes the following form: "***Based on the article, please refine the news caption provided for the image, taking into account its associated confidence score. In the article, there is a confidence score in parentheses after each sentence, indicating the correlation between the sentence and the image.*** + $I$ + ***Caption:*** $y'_{news}(\hat{f})$ + ***Article:*** $s_n(f_n)$". This methodology significantly mitigates error accumulation within the LLaVA model when processing complex inputs, thereby further refining the quality of generated news captions and ensuring a higher degree of accuracy.

$$\hat{y}_{news} = \text{LLaVA}(I, y'_{news}, \hat{f}, s_n, f_n), \hat{f} < \mu \quad (8)$$

where $\hat{y}_{news}$ denotes the final output news caption.

# 4 EXPERIMENT

## 4.1 Experimental Setup

**Datasets.** We evaluate the proposed method on two news captioning datasets: GoodNews [2] and NYTimes800k [28]. The data in these two datasets is collected using The New York Times public API [1]. GoodNews dataset comprises 257,033 news articles and 462,642 images, while the larger NYTimes800k dataset consists of 444,914 news articles and 792,971 images. We follow the split methodology described in [28] to partition the datasets into training, validation, and test sets. The GoodNews dataset includes 421,256 training samples, 18,335 validation samples, and 23,051 test samples, while the

NYTimes800k dataset comprises 763,217 training samples, 7,777 validation samples, and 21,977 test samples.

**Metrics.** We employ four evaluation metrics, namely BLEU-4 [18], ROUGE [6], METEOR [1], and CIDEr [29], to assess the quality of the generated news captions. The CIDEr metric is particularly well-suited for the news captioning task as it places greater emphasis on uncommon vocabulary compared to other evaluation metrics. Furthermore, we utilize the SpaCy toolkit to identify named entities in both the generated news captions and their corresponding ground truth captions, and we report the precision and recall scores for named entities in the generated news captions.

**Training Details.** Based on the PyTorch framework [19], we fine-tune the entire CLIP model [21] on the two news captioning datasets. Specifically, we treat the image-caption pairs in the training set as standard-positive samples. Then, we freeze all the parameters of the pre-trained MPNet [2] as the teacher model and employ it to further filter potential-positive samples, where the value of $\eta$ in Eq. 2 is set to 0.7. We carry out this process on one RTX 3090Ti GPU, with 3 epochs. The learning rate is $5 \times 10^{-6}$, and the batch size is 50. Additionally, we perform two rounds of fine-tuning on the LLaVA model using the LoRA strategy with rank of 128, one for confidence-aware prompt enhancement and the other for confidence-feedback prompt enhancement. For the former, considering the substantial computational resources and time required for processing and training large-scale datasets, we randomly sample a portion of the data from the training sets of the two news caption datasets to strike a balance between research efficiency and model performance. Specifically, we randomly sample 10% of the training data from GoodNews and 20% from NYTimes800k to achieve better performance. Experiments demonstrate that we achieve superior results using less training data. For the latter, we utilize the validation sets of GoodNews and NYTimes800k for fine-tuning LLaVA. The $\mu$ in Eq. 8 is set to 0.5 for GoodNews and 0.55 for NYTimes800k. The two fine-tuning sessions of the LLaVA model used the same parameter configuration, and we complete this work using the DeepSpeed framework [23]. We conduct the entire model fine-tuning process on two RTX 3090Ti GPUs. The learning rate is $2 \times 10^{-4}$, and the batch size is 16. For the GoodNews dataset, the two stages of fine-tuning require 15 hours and 7 hours, respectively, while for the NYTimes800k dataset, the corresponding times are 53 hours and 3 hours.

## 4.2 Comparison with the State-of-The-Art

We compare the proposed method with state-of-the-art news captioning methods, including template-based methods [2, 22] and end-to-end methods [20, 28, 35, 40, 45]. Additionally, we also compare our method with the baseline, which is obtained by fine-tuning the LLaVA model directly using images and news article. As shown in Table 1, our method demonstrates superior performance on both the GoodNews and NYTimes800k datasets, achieving state-of-the-art results on the majority of evaluation metrics. We have the following three observation:

**1)** Our method achieves state-of-the-art performance on most metrics for the news captioning task. For the GoodNews dataset, the proposed method achieves the best results across all general image captioning metrics, including BLEU-4, ROUGE, METEOR, and

---

[1] https://developer.nytimes.com/apis

[2] https://huggingface.co/sentence-transformers/all-mpnet-base-v2

**Table 1: Evaluation results on goodnews and nytimes800k datasets. All values are reported as percentages.**

| | Method | Pre-Train Data | BLEU-4 | METEOR | ROUGE | CIDEr | Named Entities P | R |
|---|---|---|---|---|---|---|---|---|
| GoodNews | Ramisa et al. [22] | Full Train | 0.89 | 4.45 | 12.09 | 15.35 | - | - |
| | Biten et al. [2] | Full Train | 0.89 | 4.37 | 12.20 | 13.10 | 8.23 | 6.06 |
| | Tell [28] | Full Train | 6.05 | 10.30 | 21.40 | 53.80 | 22.20 | 18.70 |
| | JoGANIC [35] | Full Train | 6.83 | 11.25 | 23.05 | 61.22 | 26.87 | 22.05 |
| | Zhou et al. [45] | Full Train | 6.30 | - | 22.40 | 60.30 | 24.20 | 20.90 |
| | NewsMEP [40] | Full Train | 8.30 | 12.23 | 23.17 | 63.99 | 23.43 | 23.24 |
| | Qu et al. [20] | Full Train | 7.91 | 11.67 | 22.61 | 69.11 | 24.38 | 23.56 |
| | **Baseline** | 10% Train | 8.31 | 12.70 | 25.75 | 81.28 | 28.96 | 25.76 |
| | **Ours** | 10% Train, Full Val | **8.49** | **12.88** | **26.22** | **83.52** | **30.19** | **26.57** |
| NYTimes800k | Tell [28] | Full Train | 6.30 | 10.30 | 21.70 | 54.40 | 24.60 | 22.20 |
| | JoGANIC [35] | Full Train | 6.79 | 10.93 | 22.80 | 59.42 | 28.63 | 24.49 |
| | Zhou et al. [45] | Full Train | 7.00 | - | 22.90 | 63.60 | 29.80 | 25.90 |
| | NewsMEP [40] | Full Train | **9.57** | 13.02 | 23.62 | 65.85 | 26.61 | 28.57 |
| | Qu et al. [20] | Full Train | 8.81 | 12.00 | 23.02 | 70.83 | 27.57 | 27.16 |
| | **Baseline** | 20% Train | 8.59 | 12.74 | 25.83 | 79.96 | 31.61 | 29.01 |
| | **Ours** | 20% Train, Full val | 9.07 | **13.17** | **26.48** | **83.72** | **32.38** | **30.08** |

CIDEr, outperforming the best competitors [40] and [20] by 0.19%, 0.65%, 3.05%, and 14.41%, respectively. For the NYTimes800k dataset, we only compare our model with a subset of well-performing models. The results demonstrate that our method achieves 13.17%, 26.48%, and 83.72% on the ROUGE, METEOR, and CIDEr metrics, respectively, surpassing all current template-based and end-to-end models. Notably, compared to the best competitor [20], we improve the CIDEr score by 12.89%. These results substantiate the effectiveness of eliminating irrelevant information from the model input and output stages.

2) Our method also shows better performance on named entity recognition. In this work, we employ knowledge distillation strategy to fine-tune an image-text matching module, which adapt to the news domain to identify sentences containing key information in news articles. Thanks to this image-text matching module, our named entity precision reaches 30.19% on the GoodNews dataset and 32.38% on the NYTimes800k dataset, improving by 3.32% and 2.58% respectively over the best competitors [35] and [45]. Meanwhile, our named entity recall rate improves by 3.01% on GoodNews and 1.51% on NYTimes800k compared to the best competitors [20] and [40], respectively.

3) Benefiting from the image-text matching guided feedback learning strategy, our method achieves significant improvements over the baseline. As shown in Table 1, our method outperforms the baseline across all metrics on both news captioning datasets. Specifically, for the CIDEr metric, our method improves upon the baseline by 2.24% on the GoodNews and 3.76% on the NYTimes800k. This result clearly demonstrates the practicality of our adopted strategy of feedback learning.

## 4.3 Ablative Analysis

We explore the effectiveness of our method by answering the following four key questions. **Q1**: Does the confidence-aware prompt strategy reduce input noise? **Q2**: Does the confidence-feedback

prompt strategy reduce output noise? **Q3**: Is it necessary to use knowledge distillation technology to create a specialized image-text matching module for the news domain? **Q4**: What kind of impact would using a multi round feedback iteration mechanism have?

**Effectiveness of the confidence-aware prompt strategy (Q1).** By designing a confidence-aware fine-grained prompt, we aim to enhance the LLaVA model's ability to recognize key information in long articles, thereby improving its accuracy in handling complex visual-language tasks. To validate the effectiveness of this strategy, we design a variant Non-Confidence-Aware, which intentionally omit the crucial step of confidence-prompt learning. This variant directly conduct feedback learning on the news captions generated by the baseline method, in order to explore the specific impact of input noise for news captioning. As shown in Table 2, our method improve named entity accuracy by 0.55% on GoodNews and 1.96% on NYTimes800k comparing with Non-Confidence-Aware. This result proves that our method can more accurately capture key information in news articles.

**Effectiveness of the confidence-feedback prompt strategy (Q2).** Our method introduces a feedback learning mechanism to reduce the effect of input noise for news captioner. In order to investigate its specific effects, we design a variant named Non-Confidence-Feedback. Specifically, we remove the feedback learning mechanism and directly evaluate the experimental effect of introducing confidence-aware prompt strategy. The results are shown in Table 2. Our method achieves superior performance on both news captioning datasets, exhibiting the effectiveness of the confidence-feedback prompt strategy in improving the relevance of the output caption with news image.

**Effectiveness of the image-text matching module (Q3).** Our method employ knowledge distillation strategy to fine-tune CLIP, which aims to develop a specialized image-text matching module for news content. To verify its necessity, we first employ zero-shot and fine-tuned CLIP to calculate the confidence scores between images and sentences of news article in testing data of both dataset,

**Table 2: Comparison between our method and two variants. All values are reported as percentages.**

| | Method | BLEU-4 | METEOR | ROUGE | CIDEr | Named Entities P | R |
|---|---|---|---|---|---|---|---|
| GoodNews | Non-Confidence-Aware | 8.47 | 12.83 | 26.09 | **83.52** | 29.64 | 26.08 |
| | Non-Confidence-Feedback | 8.45 | 12.82 | 25.83 | 81.09 | 28.60 | 26.23 |
| | Ours | **8.49** | **12.88** | **26.22** | **83.52** | **30.19** | **26.57** |
| NYTimes800k | Non-Confidence-Aware | 9.01 | 13.06 | 25.97 | 79.13 | 30.42 | 29.54 |
| | Non-Confidence-Feedback | 9.05 | 13.00 | 26.10 | 80.60 | 31.36 | 29.23 |
| | Ours | **9.07** | **13.17** | **26.48** | **83.72** | **32.38** | **30.08** |

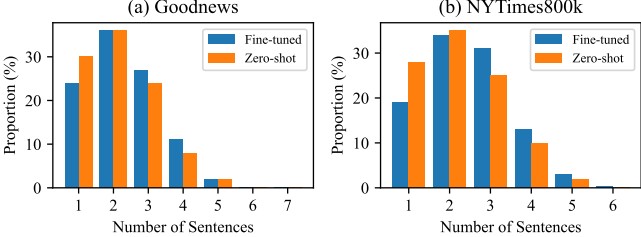

**Figure 4: The distribution chart of articles based on the varying counts of sentences that scored above 0.1. Each article's sentences are scored using both the Fine-tuned CLIP and the Raw CLIP models. (a) for Goodnews, (b) for NYTimes.**

respectively. According to the preliminary results, we observe that when a sentence in a news article achieves a confidence score of over 0.1, two kinds of CLIP model consider it as a relatively high rated sentence, and these sentences typically contain rich key named entity information. The detailed statistical analysis results of the rated news sentences are shown in Fig. 4, which reveal an interesting phenomenon: despite each news article containing 17 sentences in average, fewer than four sentences attain a relevance score exceeding 0.1 in 98% of the sample articles, which support our conclusion that sentences with scores exceeding 0.1 are key sentences identified by the CLIP model.

Furthermore, we quantitatively evaluate the image-text matching ability of zero-shot and fine-tuned CLIP in news domain. Firstly, we define that the correct key sentences in news article contain the same named entity as the ground truth caption. Then, we evaluate the ability of two kinds of CLIP model to distinguish key sentences. As shown in Table 3, the identified key sentences with confidence scores exceeding 0.1 almost encompass all named entities present in ground truth captions for both CLIP model. Nevertheless, the fine-tuned CLIP significantly surpasses the zero-shot CLIP in identifying news sentences containing key named entities, which confirms the superior capability of our module in comprehensively capturing image-related key information from news article.

**Impact of multi round iteration mechanism (Q4).** For the proposed feedback learning strategy, we aims to investigate the impact of multiple iterations for the model's named entity recognition in output news caption, involving the variant of the number of iterations. We report the trends in the accuracy and recall of named entities under different iteration rounds in Fig. 5. The results show that comparing with discarding the feedback learning process (0

**Table 3: Comparison of zero-shot CLIP and fine-tuned CLIP. Accuracy refers to the ratio of key sentences identified that contain named entities in GT captions, and the count refers to the number of correctly identified key sentences.**

| | | Accuracy | Count |
|---|---|---|---|
| GoodNews | Zero-shot CLIP | 100% | 49,770 |
| | Fine-tuned CLIP | | 53,271 |
| NYTimes800k | Zero-shot CLIP | 100% | 48,655 |
| | Fine-tuned CLIP | | 54,322 |

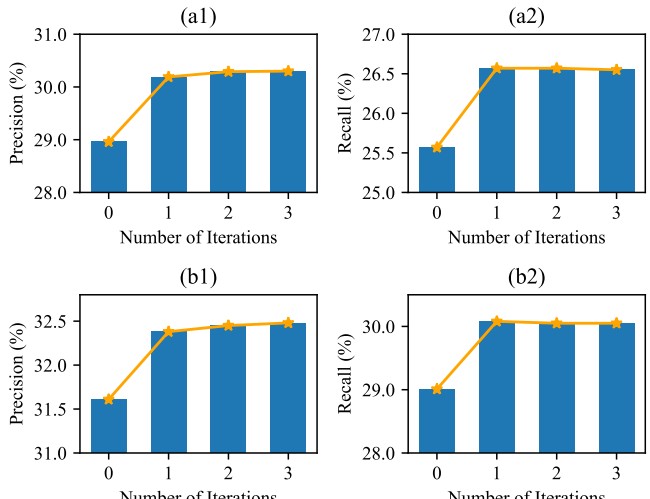

**Figure 5: The impact of the number of iterations in feedback learning on the accuracy and recall of named entities. (a1)(a2) refer to GoodNews. (b1)(b2) refer to NYTimes800k**

iterations), our method reduces the interference of information noise, thereby significantly improving the accurate and comprehensive of the named entities contained in the generated news captions. Besides, we also find that the augment of iterations have little influence for feedback learning. We suppose that this may contribute to the model quickly reaching a performance bottleneck after the initial iteration, once the model is able to identify key named entities well. Therefore, additional iterations do not provide enough new information to further optimize model performance

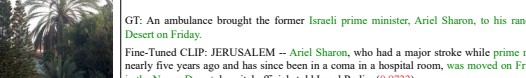

GT: The boats, the fastest and most technologically advanced in the 159-year history of the Cup, reached speeds of approximately 22 knots, or more than 25 miles an hour, in just 6 or 7 knots of wind.
Fine-Tuned CLIP: The boats, the fastest and most technologically advanced in the 159-year history of the Cup, reached speeds of approximately 22 knots, or more than 25 miles an hour, in just 6 or 7 knots of wind. (0.9209)
Zero-Shot CLIP: Spithill's early move forced a penalty as Alinghi blocked BMW Oracle's path, but Spithill then somehow stalled the trimaran, enabling Bertarelli to steer Alinghi around the marker and into the lead. (0.2874) But BMW Oracle gained ground quickly and drew even about a third of the way into the 20-nautical-mile leg. (0.5044)

GT: An ambulance brought the former Israeli prime minister, Ariel Sharon, to his ranch in the Negev Desert on Friday.
Fine-Tuned CLIP: JERUSALEM -- Ariel Sharon, who had a major stroke while prime minister of Israel nearly five years ago and has since been in a coma in a hospital room, was moved on Friday to his ranch in the Negev Desert, hospital officials told Israel Radio. (0.9722)
Zero-Shot CLIP: Lincoln Center has taken the idiosyncratic view in recent years that because classical music is the center's main business, the Lincoln Center Festival should mostly steer clear of it. (0.9233)

**Figure 6: Qualitative analysis of zero-shot CLIP and fine-tuned CLIP scores. GT refers to news captions written by professional journalists.**

## 4.4 Qualitative Analysis

We first compare the performance of the fine-tuned CLIP and zero-shot CLIP in the image-text matching task in the news domain in Fig. 6. Following **Q3** of 4.3, we consider sentences with a confidence score greater than 0.1 as key sentences selected by two CLIPs. The sentences corresponding to Fine-Tuned CLIP and Zero-Shot CLIP in the Fig. 6 are the key sentences selected by each of them. The main observation can be summarized as follows:

Cases (a) and (b) indicate that the fine-tuned CLIP model exhibits higher accuracy in identifying sentences closely related to image content in articles. In case (a), the model demonstrate its best performance by being able to directly locate the sentence that corresponds exactly to the ground truth caption "The boats, the fastest and most technologically advanced in the 159-year history of the Cup, received speeds of approximate 22 knots, or more than 25 miles an hour, in just 6 or 7 knots of wind." This sentence accurately describes the content presented by the image. In case (b), compared to zero-shot CLIP, the fine-tuned model is able to efficiently extract the sentences that best match the ground truth caption from the article, including key named entities and specific event.

Fig. 7 shows examples of news captions generated by our method, baseline method, and Tell [28]. We mark correct news sentences (i.e., high-confidence sentences) and news caption fragments in green, while incorrect news sentences and news caption fragments in red. The results show that the news captions generated by our method can capture the correct named entities and accurately reflect the image content. Specifically, we make the following three observations:

**1)** The proposed method effectively guides the model to focus on the named entity information in the key sentences of the news article through confidence-aware prompt, thereby improving the model's precision in named entity recognition. For example, in case (a), when given an image of a person being arrested, the baseline method and the Tell method incorrectly identify the person as "Mr. McCrory". In contrast, our method captures the correct named entity information by guiding the model to focus on the key news context. According to the sentence "'It shocked everybody,' said Sam Hummel, a 76-year-old retired investment adviser who was arrested ... known as 'Moral Mondays.'", the model can easily infer that the real identity of the arrested person is "Sam Hummel".

| Image | Article |
|---|---|
| 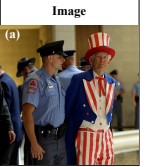 | North Carolinians Fear the End of a Middle Way(0.0) In an interview, Mr. McCrory said that critics ... responsible agenda.(0.0) With a run of ... they held their convention in Charlotte in 2012.(0.0033) "It shocked everybody," said Sam Hummel, a 76-year-old retired investment adviser who was arrested last month in Raleigh wearing an Uncle Sam costume and taking part in the protests that have come to be known as "Moral Mondays." (0.9927) North Carolina ... was almost entirely Democratic.(0.0001) ... |

**Our:** Sam Hummel, a 76-year-old retired investment adviser, was arrested last month in Raleigh, N.C., while protesting.
**Baseline:** Mr. McCrory, who was arrested last month, was charged with trespassing on the grounds of the State Capitol.
**Tell:** Gov. Pat McCrory of North Carolina, center, at a rally in Raleigh last week. He has been arrested in Raleigh since the fall of 2008.
**GT:** Sam Hummel, a 76-year-old retired investment adviser who was arrested at the "Moral Monday" protest.

| Image | Article |
|---|---|
| 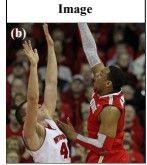 | ... last 12 meetings.(0.0) Michael Dixon added two free throws with 9.8 seconds left after an offensive foul on Tyshawn Taylor, ... it at the buzzer.(0.0004) ... Davis finished with 22 points and Jones 16.(0.0004) Damien Leonard scored a career-high 19 points for South Carolina (9-13, 1-7).(0.0) OHIO STATE 58, WISCONSIN 52 Jared Sullinger scored 24 points and had 10 rebounds as No. 3 Ohio State (20-3, 8-2 Big Ten) ended a nine-game losing streak to Wisconsin (18-6, 7-4) in Madison.(0.9985) |

**Our:** Jared Sullinger, right, had 24 points and 10 rebounds in Ohio State's victory.
**Baseline:** Marcus Denmon, right, hit three 3-pointers in the final 2 minutes 5 seconds to give Missouri a 1-point lead with 56 seconds to go.
**Tell:** Missouri's Michael Dixon shooting against Kansas' Tyshawn Taylor in the final seconds of Saturday's game.
**GT:** Jared Sullinger scored 24 for Ohio State in a victory on Saturday at Wisconsin.

**Figure 7: Qualitative results of our method on the Good-News dataset. We mark the incorrect news sentences and captions in red, while the correct news sentences and captions in green.**

**2)** Our image-text matching module can accurately locate the news sentences that match the image content in long articles, thereby guiding the model to ignore irrelevant information and generate accurate news captions. For example, in case (b), the image-text matching module identifies the correct news sentence "OHIO STATE 58, WISCONSIN 52 Jared Sullinger scored 24 ... Wisconsin (18-6, 7-4) in Madison.", and assigns it a high confidence score of 0.9985. This confidence prompt helps the model better understand the association between text and image, thereby generating the accurate description "Jared Sullinger, right, had 24 points and 10 rebounds in Ohio State's victory." In contrast, due to the lack of confidence score prompting, the baseline and Tell methods locate the wrong news context, leading to generate incorrect captions.

## 5 CONCLUSION

In this work, we propose the novel cross-modal coherence-enhanced feedback prompting method for news captioning, which effectively reduces the impact of input and output noise on captioner. Specifically, we first adopt cross-modal knowledge distillation technology to construct a specialized image-text matching module for news domain. Next, we utilize this module to design a confidence-aware prompt mechanism and fine-tune the LLaVA model using LoRa technology to address input noise issues. We also adopt a feedback learning strategy to correct the generated news caption through iterative optimization to improve its accuracy. The experimental results on two large news caption datasets validate the effectiveness of our method. In the future, we also plan to explore the use of reinforcement learning methods to optimize feedback learning strategy, enabling models to dynamically adjust model's generation strategy to adapt to various noise conditions.

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
