# OpenReview forum: "Cross-Modal Coherence-Enhanced Feedback Prompting for News Captioning"
_acmmm.org/ACMMM/2024/Conference — MM2024 Poster_

### Official Review · Reviewer_vGUW · 2024-05-20

**Rating:** 5
**Confidence:** 4

**Summary:**

This is a rather interesting paper that introduces a feedback mechanism in the process of solving text noise interference. Through interactive refinement of caption generation, this paper provides a novel approach to the field of news caption generation. The entire article revolves around addressing the problem of frequent noise information unrelated to images in articles. It proposes the use of a refined CLIP model to calculate the similarities between sentences in articles and captions and fine-tunes the LLaVA model for caption generation accordingly. This helps mitigate the impact of text noise during input. Additionally, during output, the similarity between the generated caption and the image is calculated using the refined CLIP model modified before adjusting inappropriate outputs. The entire design logic is highly coherent and supported by a large number of experiments demonstrating the effectiveness of the proposed method. Overall, the article is innovative and contains substantial work.

**Strengths:**

This is an inspiring work, where the use of similarity scores throughout all designed modules in the experiment ensures a highly logical flow in the entire paper. The innovative idea of continuously adjusting the output through feedback interaction, as mentioned in the article, opens up new avenues for the news caption task. Additionally, in the design section of each module, the reasons for the design and the problems to be addressed are clearly articulated. When discussing existing issues, the article is vividly illustrated, making it more visual and easier to comprehend.

**Limitations:**

Although it is good to see the paper as a whole, there still exist some minor drawbacks in this work:
1. Why was MPNet chosen as the teacher model for distillation rather than others?
2. In cross-modal knowledge distillation methods, an ablation study should be conducted on parameter values for the threshold used to filter out news sentences with low matching scores to the true caption.
3. The designed prompt lacks an explicit indication of the numerical range for similarity scores, such as the range from 0 to 1, where values closer to 1 indicate higher similarity.
4. In Figure 6, the colors used for case a and case b are not sufficiently distinct and blend into the background of the image, making them difficult to differentiate.

**Suitability:**

3

---

### Official Review · Reviewer_jC9P · 2024-05-21

**Rating:** 3
**Confidence:** 3

**Summary:**

This paper presents an intricate pipeline for generating news captions. Specifically, it leverages a pre-trained Large Language Model, MPNet, as a teacher model to guide CLIP in acquiring news-specific image-text matching capabilities. Subsequently, it utilizes refined CLIP to evaluate the matching degree between generated news sentences and corresponding images, thereby facilitating feedback learning mechanisms to fine-tune LLaVA. Experimental results show that LLaVA fine-tuned using this approach outperforms direct fine-tuning with (image-news-article) data.

**Strengths:**

In terms of experimental results, the proposed method performs better than the compared methods

**Limitations:**

The primary limitation lies in the lack of clarity regarding the scientific contribution of the paper. While the paper proposes a strategy to fine-tune LLaVA, claiming better performance with fewer data, it fails to establish the necessity of this method. What is the zero-shot performance of LLaVA? How does fine-tuning LLaVA with the entire dataset, as suggested by the baseline in the paper, impact its performance? If LLaVA performs sufficiently well under these conditions, does the proposed method become redundant? Thus, the experimental comparisons are also inadequate.

**Suitability:**

3

---

### Official Review · Reviewer_SeBQ · 2024-05-24

**Rating:** 4
**Confidence:** 3

**Summary:**

This paper studies the task of News Image Captioning (NIC), with a focus on the use of large multimodal models to enhance the precision of news captioning by effectively mitigating the impact of irrelevant details. The study identifies some limitations of existing methods that fail to minimize the impact of irrelevant context without losing pertinent information. To address this, the authors propose a well-designed two-stage finetuning strategy incorporating confidence-aware prompts and feedback. Particularly, this approach is based on a News-specific image-text matching module trained by cross-modal knowledge distillation to estimate the confidence score. Experiments demonstrate the effectiveness of the proposed method, significantly advancing the SoA on two popular benchmarks for this task and qualitatively improving caption quality.

**Strengths:**

1. This is one of the first attempts to utilize large multimodal models for (NIC), given their strong performance across a wide range of multimodal tasks.
2. The paper presents a carefully designed two-stage finetuning strategy for LlaVa with confidence-aware prompts and feedback learning. To achieve this, the authors also propose to finetune CLIP to adapt it to News domains by levegaring cross-modal knowledge distillation with MPNET as the teacher, in order to estimate the confidence. This approach significantly outperforms previous state-of-the-art methods on GoodNews and NYTimesork800k while using significantly less data.
3. Overall, the paper is well-written and easy to follow.

**Limitations:**

1. In my opinion, most of the improvement compared to previous works comes from the use of LlaVa. The finetuned model (baseline) can already achieve performance close to the proposed approach (Table 1), which is expected. As a result, the contribution of the paper is somewhat limited.
2. The proposed method using CLIP finetuned with cross-modal knowledge distillation is interesting and addresses the limitation of prior work that does not consider the relevance of contextual information to the image. To better demonstrate its effectiveness, it is suggested to apply this module to existing methods to see if it results in improvements, as this issue might not be observed in the finetuned LlaVa.
3. In the Ablation Study, while confidence-feedback show its effectiveness, the benefit of confidence-aware prompt is unclear. The improvement from incorporating this strategy is marginal in almost all metrics, except for a +4 points CiDER in the case of NYTimes800k only (Table 2). Moreover, adding this to the baseline do not yield significant improvements either (comparing row 2 in Table 2 and the baseline in Table 1, CiDER score even decreases in GoodNews).
4. Also, it would be helpful to understand the benefits of confidence-feedback with an example showing how the confidence score changes after the first round of feedback.

**Suitability:**

3

---

### Official Review · Reviewer_admq · 2024-05-26

**Rating:** 3
**Confidence:** 3

**Summary:**

This paper proposes a method called "Cross-Modal Consistency Enhanced Feedback Cueing", which aims to improve the description generation process for news images by ensuring that the generated descriptions are closely aligned with the content of the news images. The method specifically addresses a common problem in news image word matching tasks, where news articles contain a large amount of extended information that is not directly related to the content of the image, which may result in a mismatch between the generated description and the visual content.

**Strengths:**

1. The paper presents a methodology for tuning pre-trained models using knowledge distillation techniques, which not only enhances the ability of the models to process domain-specific data, but also paves the way for more targeted and efficient processing in this domain. The proposed module is a significant step forward in the development of domain-specific natural language processing systems, demonstrating the potential of such strategies in fine-tuning large language models for specialized applications.
2. This paper proposes a commendable solution to the problem of inaccurate headline generation often caused by mismatch between news article content and images. The proposed method effectively reduces the noise in the input data and model output, greatly improving the accuracy of the generated headlines. The design of the confidence-based cueing mechanism, coupled with the fine-tuning of the LLaVA model using LoRa technology, ensures that the model focuses on key sentences in lengthy articles. This strategic approach not only improves the relevance of the generated headlines, but also demonstrates the potential of adaptive learning strategies in natural language processing tasks. The integration of these techniques is particularly noteworthy and has the potential to set new standards for domain-specific machine learning models.
3. The paper has undergone extensive experimental validation to substantiate the effectiveness of the proposed method. The use of two large-scale news headline datasets, GoodNews and NYTimes800k, serves as compelling evidence of the model's superior performance. Furthermore, a detailed analysis of the results not only corroborates the model's improved capabilities but also highlights an enhancement in named entity recognition. This is particularly noteworthy given the nuanced approach towards refining the LLaVA model through a confidence-based prompt mechanism and LoRa technology fine-tuning, which have collectively led to enhanced accuracy in generating relevant headlines.

**Limitations:**

1. Weak innovative persuasion，while the results look promising, the persuasiveness of the innovation leaves much to be desired. Through ablation experiments, the two main innovations designed by the authors provide minimal enhancement to the results. It seems possible to attribute the model's performance primarily to the MPNet, CLIP, and LLaVA models. In order to strengthen the paper, the authors should split the modules for further experiments and detailed descriptions. to demonstrate their importance and unique contribution to the final performance improvement. Comparing the results of these ablations can provide a clearer picture of what factors are critical to realizing the reported gains.
2. The analysis of the results seems insufficient. The authors should provide a more rigorous statistical analysis to substantiate their claims about model performance and improvements over existing methods. Additionally, they should discuss the impact of hyperparameters and data split strategies on the reported outcomes. A comparison with state-of-the-art baselines would also strengthen the paper by demonstrating the superiority of the proposed method. Furthermore, an analysis of failure cases and limitations is necessary to offer a balanced view of the model's capabilities and areas for potential enhancement. Finally, discussing how these results translate to real-world applications would improve the paper's relevance and impact.
3. The discussion on the long-term stability and robustness of the model could be strengthened. The authors should evaluate how the model performs under conditions of noise in the data or dynamic changes in the environment. This is crucial for real-world applications where the data may not be perfectly clean or consistent. Without addressing these aspects, the practical implications and limitations of the method remain unclear. A thorough analysis of the model's resilience to such challenges would improve confidence in its applicability and reliability for long-term use.

**Suitability:**

3

---

### Meta-Review · Area_Chair_z7vB · 2024-07-02

**Recommendation:** Accept (Poster)
**Confidence:** 4

**Metareview:**

This paper proposes a cross-modal coherence-enhanced feedback prompting method, to better assess relevant visual elements for the task of news image captioning.

In terms of its strengths, reviewers point out:
- The proposed two-stage methodology to fine-tune large V&L models (e.g. LLaVA) to the news image captioning task (admq, SeBQ, vGUW).
- Overall effectiveness of the proposed approach (jC9P).
- The paper is well-written and easy to follow (SeQb).

As to its weaknesses highlights, reviewers mention:
- The effectiveness of the proposed approach components, according to the conducted ablation study, is unclear (admq, SeBQ).
- The conducted experiments lack a statistical analysis of the results (admq).
- Performance of the baseline model was not evaluated (jC9P). This was addressed in the rebuttal.

The author's rebuttal addressed some of the reviewers' concerns, but not all. The authors attempted to clarify the reviewer's admq concerns, but the reviewer did not provide his final score.

As such, to sum up, the paper got one Borderline Reject, two Borderline Accepts, and one Weak Accept. These scores, overall, tend toward acceptance. Therefore, and recognizing this paper merits, I suggest this work to be accepted as Poster if there is space, provided that authors include the clarifications discussed in their rebuttal in the final manuscript.